



# The underappreciated role of transboundary pollution in future air quality and health improvements in China

Jun-Wei Xu[1], Jintai Lin[1]*, Dan Tong[2], Lulu Chen[1]

[1]Laboratory for Climate and Ocean–Atmosphere Studies, Department of Atmospheric and Oceanic Sciences, School of Physics, Peking University, Beijing, China

[2]Department of Earth System Science, Ministry of Education Key Laboratory for Earth System Modelling, Tsinghua University, Beijing, China

Correspondence: Jintai Lin (linjt@pku.edu.cn).

**Abstract**

Studies assessing the achievability of future air quality goals in China have focused on the role of reducing China's domestic emissions, yet the influence of transboundary pollution of foreign origins has been largely underappreciated. Here, we assess the extent to which future changes in foreign transboundary pollution would affect the achievability of air quality goals in 2030 and 2060 for China. We find that adopting the low-carbon instead of the fossil fuel-intensive pathway in foreign countries would avoid millions of Chinese people from being exposed to $PM_{2.5}$ concentrations above China's air quality standard level (35 µg m$^{-3}$) in 2030 and the World Health Organization Air Quality Guideline (5 µg m$^{-3}$) in 2060. China adopting the carbon-neutral pathway rather than its current pathway would also be helpful to reduce transboundary $PM_{2.5}$ produced from the chemical interactions between foreign-transported and locally-emitted pollutants. In 2060, adopting a low-carbon pathway in China and foreign countries coincidently would avoid 63% of transboundary pollution and 386,000 associated premature deaths in China, relative to adopting a fossil fuel-intensive pathway in both regions. Thus, the influence of transboundary pollution should be carefully considered when making future air quality expectations and pollution mitigation strategies.

## 1. Introduction

Long-term exposure to ambient fine particulate matter ($PM_{2.5}$, particulate matter smaller than 2.5 µm in aerodynamic diameter) is the largest environmental risk factor for human health, with an estimated 4.1 million attributable deaths worldwide (7.3% of the number of global deaths in 2019; Murray et al., 2020). Countries have taken diverse actions to improve air quality and public health, including setting ambitious future air quality and/or climate goals. The European Commission set the 2030 air quality goal as reducing the number of $PM_{2.5}$-attributable premature deaths by at least 55% compared with 2005 levels, equivalent to reducing $PM_{2.5}$ concentrations to below 10 µg m$^{-3}$ EU-wide (European Commission, 2022). The United States aims for a 50-52% reduction of greenhouse gas emissions relative to 2005 levels by 2030, which would avoid tens of thousands of premature deaths associated with $PM_{2.5}$ as a co-benefit (Burtraw et al., 2021). However, the achievability of air quality goals or the extent of health co-benefits of climate strategies is subject to large uncertainties as it is affected by a wide range of factors such as the domestic and global socioeconomic development pathways, energy choices, and air pollution control measures (Cheng et al., 2021b; O'Neill et al., 2020; Rao et al., 2017).



China is a key country to examine factors affecting the achievability of future air quality goals.
On the one hand, China suffers from serious air pollution and adverse health effects, with more
than a quarter of the world's total $PM_{2.5}$-associated premature deaths in 2015 estimated to occur
in China (Zhang et al., 2019). Despite remarkable achievements of the 5-year Clean Air Action
since 2013 (Zhang et al., 2019; Zheng et al., 2018a), annual mean $PM_{2.5}$ concentration in China
still exceeded the newly revised World Health Organization (WHO) Air Quality Guideline
(WHO, 2021; AQG; 5 µg m$^{-3}$) by 8 times by the end of the Clean Air Action in 2017. On the
other hand, China has set the most challenging air quality and climate goals among all the
developing countries in the world (Ascensão et al., 2018). In 2018, the Chinese government
proposed the roadmap for The Beautiful China Initiative (The State Council of the People's
Republic of China, 2016), which requires all cities to achieve the national ambient air quality
standards (NAAQS, 35 µg m$^{-3}$) between 2030 and 2035 (The State Council of the People's
Republic of China, 2016). In 2019, China further announced an ambitious climate commitment
to achieve carbon neutrality by 2060. These policies have been regarded as interim steps towards
the WHO AQG. Therefore, investigating potential pathways and factors shaping the future air
quality in China could provide an excellent reference for other countries facing the dual
challenges of economic development and environmental protection.
A number of studies seek to answer whether the air quality goals in China can be achieved in
the future. These studies have been primarily focused on potential mitigation pathways to reduce
China's domestic emissions. Cheng et al. (2021b) found that the nation's current emission
control measures could reduce China's $PM_{2.5}$ levels to below 30 µg m$^{-3}$ by 2030, yet the benefits
of such measures would be mostly exhausted by then (Cheng et al., 2021b; Xing et al., 2020).
Xing et al. (2020) and Tang et al. (2022) suggested that the co-benefits of climate targets alone
(even the most ambitious target of 1.5 ℃ limit in global warming) were not able to help China
achieve the 35 µg m$^{-3}$ target by 2035. Instead, Cheng et al. (2021b) proposed that a combination
of stringent clean air policies and ambitious climate targets (i.e., carbon neutrality by 2060 or a
global warming limit of 1.5 ℃) could successfully reduce $PM_{2.5}$ to below 35 µg m$^{-3}$ by 2030 and
to below 10 µg m$^{-3}$ (WHO interim target 4) by 2060. This would require a fundamental
transformation of China's economic-environmental development pathway by phasing out
polluting industries and moving towards renewable energy while implementing strict end-of-pipe
emission control.
However, these previous studies have largely neglected how the future changes in foreign
transboundary pollution would affect air quality in China, likely due to the perception that the
relatively short lifetime of $PM_{2.5}$ (a few days) does not permit long-distance transport. This
negligence would put into question the confidence of their estimated achievability of air quality
goals in China. Due to large uncertainties in the future socioeconomic development pathways,
environmental commitments, financial supports and technology capabilities, future emissions
from China's neighboring countries might be highly variable, leading to different levels of
transboundary impacts to China's air quality. For example, emissions in South Asia, Central Asia
and Southeast Asia have been estimated to increase in the future by various projections
(International Energy Agency, 2021; Koplitz et al., 2017), due to their rapid-economic growth
and a lack of clear commitments on energy choices, climate actions and air pollution control
efforts. In this case, transboundary pollution from neighboring countries to China could
potentially become increasingly important to affect the achievability of air quality goals in





China. Alternatively, these surrounding countries may undergo a sustainable development
pathway, facilitated in part by external financial aids and technology supports, leading to
lowered transboundary pollution to China. Thus, the different prospects of transboundary
pollution could be a significant yet highly uncertain factor in the achievement of future air
quality goals in China.
In addition, the mechanism of transboundary pollution poses additional complexity to its
impacts on China's air quality. Foreign emissions affect air quality in China through direct
transboundary transport in the atmosphere. Moreover, portions of foreign-transported pollution
can also interact chemically with China's locally emitted pollutants, leading to additional
transboundary effects (e.g., through formation of nitrate and ammonium; Xu et al., 2022). The
chemical interactions mean that the extent of transboundary pollution will depend on emission
changes in China as well. Therefore, the influence of transboundary pollution on China's air
quality in the future is a complex result of future emissions in China and in foreign countries,
along with their interactions. However, whether the changes in transboundary pollution via direct
transport and chemical interactions could affect the achievement of future air quality goals in
China and to what extent the influence could be remain poorly understood.
Here, we assess the potential influences that transboundary pollution could make on the
achievement of future air quality goals in China, considering the changes in direct pollution
transport and in the interactions between transported and China's locally emitted pollution. We
regard the air quality goal stated in The Beautiful China Initiative, which is that all cities have
annual mean $PM_{2.5}$ concentrations below 35 µg m$^{-3}$ between 2030~2035 (The State Council of
the People's Republic of China, 2016), as China's 2030 air quality goal. We also regard the
WHO AQG (annual mean $PM_{2.5}$ below 5 µg m$^{-3}$) as China's 2060 air quality goal and discuss the
likelihood of the achievement under currently proposed development pathways in China and
foreign countries. As detailed in Method, given the large uncertainties on future emissions of a
country (Rao et al., 2017), we consider three anthropogenic emission scenarios (low: SSP119,
medium: SSP245, and high: SSP370; O'Neill et al., 2014) for foreign countries and two
anthropogenic emission scenarios (current-policy and carbon-neutral; Cheng et al., 2021b; Tong
et al., 2020) for China (Table 1), so as to understand transboundary pollution in China from the
present (2015) to the future (2030 and 2060) under a wide range of plausible futures. We do not
consider the effects of physical climate change (e.g., temperature and precipitation) and natural
emissions of pollutants on future transboundary pollution, since their impacts on $PM_{2.5}$ are
smaller than those of anthropogenic emissions of pollutants (Hong et al., 2019; Jiang et al., 2013;
Silva et al., 2017). For each combination of foreign and Chinese anthropogenic emission
scenarios, we simulate $PM_{2.5}$ concentrations at a 0.5° x 0.625° resolution using a chemical
transport model (GEOS-Chem; http://www.geos-chem.org) that can represent the complex
pollutant emissions and chemical reactions across a large spatial domain. Then, we correct the
systematic bias in the model simulated $PM_{2.5}$ concentrations using a large set of ground-based
observations of $PM_{2.5}$ in China; the same correction factor is applied to all present and future
concentrations. Finally, we quantify the transboundary impacts on Chinese $PM_{2.5}$ under each
emission scenario. We further quantify the corresponding transboundary impacts on public
health in China, measured by premature deaths, using socio-demographic projections consistent
with SSPs and the state-of-the-art concentration–response relationships (the GEMM model;
Burnett et al., 2018).



## 2. Method

In this study, we use a set of data and models to investigate future air quality and health burden in China. Projected air pollutant emissions for foreign countries under SSP-RCP scenarios are obtained from the International Institute for Applied Systems Analysis (IIASA; Rao et al., 2017; Riahi et al., 2017) with updates on base year emissions and the harmonization year in this study. Projected air pollutant emissions for China are developed by Tong et al. (2020) and Cheng et al. (2021b). Ambient $PM_{2.5}$ concentrations under each scenario are simulated by the GEOS-Chem chemical transport model (http://www.geos-chem.org) and further corrected for systematic bias by a suite of ground-based observations. $PM_{2.5}$-associated mortalities are calculated using the Global Exposure Mortality Model (GEMM; Burnett et al., 2018).

### 2.1 Scenarios of future anthropogenic air pollutant emissions

Future pollutant emission outcome is a cumulative result of a range of variables including socio-economic development, technological change, efficiency improvements and policies directed at pollution control as well as alternative concerns including climate change, energy access, and agricultural production (Rao et al., 2017). The Shared Socioeconomic Pathways (SSP) includes 5 five distinctly different pathways about how the future might unfold in terms of major socioeconomic, demographic, technological, lifestyle, policy and institutional trends (O'Neill et al., 2014; van Vuuren et al., 2017): SSP1 - sustainability, SSP2 - middle-of-the-road, SSP3 - regional rivalry, SSP4 - inequality, SSP5 - fossil fuel development. An assumption about the degree of air pollution control (strong, medium or weak) is included on top of the baseline pathway. Weak air pollution controls occur in SSP3 and SSP4, with medium controls in SSP2 and strong controls in SSP1 and SSP5 (Turnock et al., 2020). However, SSP scenarios do not include explicit climate policies (O'Neill et al., 2020). Instead, the Representative Concentration Pathways (RCPs) generate climate projections targeting at a range of climate forcing levels, such as 1.9 W m$^2$ (1.5 ℃ warming), 4.5 W m$^2$ (3 ℃) and 7.0 W m$^2$ (4 ℃) in 2100. Thus, the combination of SSP-RCP scenario framework depicts societal and climate futures in parallel and explores plausible futures of human activities, the changing climate and emissions (O'Neill et al., 2020).

Here, we use anthropogenic aerosol emissions projected under SSP-RCP scenarios for foreign countries. We select 3 scenarios to represent low, middle and high air pollutant emissions in plausible futures: SSP119 (a sustainable development pathway targeting at a rise of the global mean surface temperature below 1.5 ℃ from the pre-industrial levels by the end of the century), SSP245 (a business-as-usual development pathway with 3 ℃ warming), SSP370 (a regional-rivalry development pathway with 4 ℃ warming). Table 1 summarizes the scenario settings in more detail.

The original SSP-RCP future anthropogenic emissions are harmonized to match anthropogenic emissions from the Community Emissions Data System (CEDS; Hoesly et al., 2018) for 2015, so that the resulting future trajectories provide a smooth transition from the historical emissions (Gidden et al., 2019). Here, we update the harmonization year from 2015 to the most recent year 2019 in CEDS historical emissions. We also use the most recently developed CEDS emissions version 2 (https://data.pnnl.gov/dataset/CEDS-4-21-21) to harmonize the future SSP-RCP



emission projections as the new emissions can better represent the historical trend of pollutant
emissions
(https://github.com/JGCRI/CEDS/blob/master/documentation/Version_comparison_figures_v_2
021_04_21_vs_v_2016_07_16(CMIP6).pdf). The final future SSP-RCP emissions for foreign
countries used in this study are presented in Fig. S1a.
Future scenarios of anthropogenic pollutant emissions for China are developed by Tong et al.
(2020) and Cheng et al. (2021b).  Briefly, we select two plausible scenarios: the current-policy
scenario and the carbon-neutral scenario. The current-policy scenario seeks to achieve China's
NDC pledges and the national $PM_{2.5}$ air quality goal (i.e. 35 μg m$^{-3}$) by 2030, elucidating China's
future air pollution mitigation pathway towards all the released and determined upcoming clean
air policies since 2015. The carbon-neutral scenario pursues China's carbon-neutral commitment
and the WHO's old $PM_{2.5}$ guideline (10 μg m$^{-3}$) by 2060. It implements the best available end-of-
pipe technologies and more stringent pollution control policies than the current-policy scenario.
Future anthropogenic pollutant emissions for China under these scenarios are developed by
firstly simulating China's future energy and socioeconomic evolution using the Global Change
Assessment Model (GCAM-China) and then translating into pollutant emissions by the Dynamic
Projection model for Emissions in China (DPEC; Tong et al., 2020). More details are
summarized in Table 1. The actual future air pollutant emissions for China used in this study are
presented in Fig. S1b.
**2.2 Simulations of ambient $PM_{2.5}$ concentrations**
We use the GEOS-Chem model to simulate $PM_{2.5}$ concentrations in China and other Asian
countries under each emission scenario and year. A number of previous studies have applied the
GEOS-Chem to simulate $PM_{2.5}$ concentrations and have shown consistency between
observations and model results (Choi et al., 2019; Koplitz et al., 2017; Venkataraman et al.,
2018; Zhang et al., 2017). We use the Flex-Grid capability of the GEOS-Chem classic model
v13.2.1 to simulate $PM_{2.5}$ concentrations over Asia and its adjacent areas (11° S–60° N, 30°–
150° E; covering China, Southern Asia, Northern Asia and Central Asia) at a horizontal
resolution of 0.5° latitude × 0.625° longitude with 47 vertical levels between the surface and ~
0.01 hPa. The lowest vertical layer has a thickness of about 130 m. We regard the pollutant
concentrations in this layer as "ground-level". Detailed descriptions of the flex-grid setup can be
found at http://wiki.seas.harvard.edu/geos-chem/index.php/FlexGrid. Our Flex-Grid domain
extend the traditionally-defined nested Asia domain (11° S–55° N, 60°–150° E) in the model to
better represent the transport of anthropogenic pollutants from Central Asia to China.
Our simulations are driven by assimilated meteorological data from MERRA-2 provided by the
Global Modeling and Assimilation Office (GMAO) at NASA Goddard Space Flight Center.
Convective transport in the model is computed from the convective mass fluxes in the
meteorological archive as described by Wu et al. (2007). A non-local scheme is used to represent
vertical mixing within the planetary boundary layer (PBL), as it accounts for different states of
mixing based on the static instability (Lin and McElroy, 2010). Chemical boundary conditions
are taken from global simulations at a resolution of 2° latitude × 2.5° longitude. We spin up
every simulation for 1 month to remove the effects of initial conditions.



GEOS-Chem simulates $PM_{2.5}$ concentrations as the sum of sulfate ($SO_4^{2-}$), nitrate ($NO_3^-$),
ammonium ($NH_4^+$), organic aerosol (OA ≡ primary OA + secondary OA), black carbon (BC),
fine dust and fine sea salt component concentrations. The sulfate−nitrate−ammonium (SNA)
aerosol system is simulated following Fountoukis and Nenes (2007) and Park et al. (2004),
including heterogeneous chemistry with dinitrogen pentoxide ($N_2O_5$) uptake by aerosol, and
hydroperoxyl radical ($HO_2$) uptake by aerosol. Gas−aerosol partitioning of SNA is simulated by
the ISORROPIA II thermodynamic equilibrium scheme (Pye et al., 2009). We use a simple
scheme to represent secondary organic aerosol formation (Heald et al., 2012) and use a spatially
resolved ratio to calculate organic mass from organic aerosol concentrations (Philip et al., 2014).
Natural dust simulation follows the Mineral Dust Entrainment and Deposition (DEAD) scheme
(Fairlie et al., 2007). Sea salt aerosol simulation is described in Jaeglé et al. (2011). Dry
deposition of gases and particles follows a standard resistance-in-series scheme, with updates
from Jaeglé et al. (2011). Wet deposition is described in Liu et al. (2001), Wang et al. (2011) and
Wang et al. (2014), with updates from Luo et al. (2020) that includes a faster below-cloud
scavenging of $HNO_3$. We calculate the simulated $PM_{2.5}$ and composition concentrations at 35%
relative humidity (RH) for consistency with ground-based measurements.
Anthropogenic emissions for the base year (2015) for China are taken from the Multi-
resolution Emission Inventory (MEIC) for 2015 (Zheng et al., 2018b), and for the rest of the
world are taken from the Community Emissions Data System (CEDS) version 2 for 2015
(https://data.pnnl.gov/dataset/CEDS-4-21-21). For future simulations, anthropogenic emissions
for China and foreign countries for each scenario are described above and are specified in Table
2. Other emissions are default in GEOS-Chem and are fixed in all present and future scenarios.
Fine anthropogenic fugitive dust emissions from combustion and industrial sources for countries
except China are taken from Philip et al. (2017), and from the MEIC inventory for China.
Aircraft emissions are from the Aviation Emissions Inventory Code (AEIC) inventory (Stettler et
al., 2011). Natural emissions include lightning $NO_x$ from Murray et al. (2012); soil $NO_x$, biogenic
non-methane volatile organic carbons (NMVOCs) and sea salt from off-line emissions developed
by Weng et al. (2020); biomass burning emissions from the Global Fire Emissions Database
version 4 (GFED4; Randerson et al., 2015); volcano emissions from Fisher et al. (2011); marine
dimethyl sulfide (DMS) emissions from Breider et al. (2017); and dust emissions from the
DEAD scheme (Zender et al., 2003).
We conduct simulations for January, April, July and October, and treat the mean of the four
months as annual mean. Our meteorological fields are fixed to 2015 for all scenarios and years to
exclude the influence of climate on the results. More details of our model configurations can be
found in Table 2. We conduct two types of simulations: 1) baseline simulations (simulations with
"Base_" prefix in Table 2) that include complete anthropogenic emissions for both China and
foreign countries; 2) sensitivity simulations (simulations with "China_" prefix in Table 2) that
exclude anthropogenic emissions for foreign countries from the baseline simulation. Baseline
simulations calculate $PM_{2.5}$ concentrations in China that are driven by both Chinese and foreign
emissions, while sensitivity simulations calculate $PM_{2.5}$ concentrations in China that are driven
merely by China's domestic emissions. The impacts of transboundary pollution on China's $PM_{2.5}$
is calculated as the difference in China's $PM_{2.5}$ between a baseline simulation and a sensitivity
simulation in a specified year and under a specified Chinese emission scenario.



We correct our simulated PM$_{2.5}$ concentrations under each scenario, based on a large set of
ground-based observations of PM$_{2.5}$, because our simulated PM$_{2.5}$ concentrations are biased high
by roughly 15.7% (Fig. S2). Descriptions and data screening method of our ground-based
observations can be found in Xu et al. (2022). Further evaluations of PM$_{2.5}$ composition
concentrations in China and PM$_{2.5}$ concentrations in other Asian countries with ground-based
observations are presented in Xu et al. (2022) . We correct PM$_{2.5}$ concentrations in grids where
PM$_{2.5}$ contributed by anthropogenic emissions (anthropogenic grids) exceeds that of natural
emissions (natural grids). To isolate anthropogenic pollution-dominated grids, we conduct a
sensitivity that excludes natural emissions in China and in foreign countries. PM$_{2.5}$
concentrations from this sensitivity simulation are referred to as natural PM$_{2.5}$ concentrations.
The difference between concentrations from the Base_2015 simulation (with complete natural
and anthropogenic emissions) in Table 2 and the natural concentrations is the anthropogenic
emission-contributed concentrations (anthropogenic concentrations). Anthropogenic grids are the
grids where anthropogenic concentrations exceed natural concentrations. We scale anthropogenic
concentrations by the average ratio of observed concentrations and simulated concentrations at
each observation site that falls into an anthropogenic grid. Concentrations before and after the
correction for anthropogenic grids and all grids (including natural grids) are shown in Fig. S2.
The overestimation in both anthropogenic PM$_{2.5}$ and overall PM$_{2.5}$ concentrations is removed
after the correction.
**2.3 Health impact assessment**
We use the GEMM model (Burnett et al., 2018) to estimate premature deaths attributable to
ambient PM$_{2.5}$ exposure for noncommunicable diseases (NCDs) and lower respiratory infections
(LRIs) in China under each scenario. GEMM NCD+LRI calculates premature deaths associated
with ambient PM$_{2.5}$ exposure (M) for each population subgroup s (by age and gender) in grid g
as:
$$M_{s,g}(C_g) = B_s \times AF_s(C_g) \times P_g \tag{1}$$
where $B_s$ is the national baseline mortality rate of NCD+LRI for the exposed population
subgroup s. $AF_s(C_g)$ is the attributable fraction of NCD+LRI to PM$_{2.5}$ exposure at level $C_g$ for
population subgroup s. $P_g$ represents the total exposed population in grid g. In particular AF was
calculated as AF = (RR − 1)/RR, where RR is the relative risk of NCD+LRI attributable to
ambient PM$_{2.5}$ exposure. The dependence of RR of NCD+LRI on PM$_{2.5}$ concentrations is
calculated as

$$RR = e^{\frac{\theta \times \ln\left(\frac{z}{\alpha}+1\right)}{1+e^{\left(-\frac{z-\mu}{\nu}\right)}}}, \text{ where } z = \max(0, C_g - 2.4) \tag{2}$$
where θ, α, μ and ν are fitted parameters of PM$_{2.5}$–mortality relationships. According to the
GEMM model, the RR of NCD+LRI is calculated by age for adults aged from 25 to greater than
85 years in 5-year intervals.
In this study, we use the national baseline mortality data by age group and disease type
43 from the Global Burden of Disease Results Tool 2017 version (GBD 2017; Institute for Health



Metrics and Evaluation, 2017). For future baseline mortality, we use the age-specific baseline
mortality rates projected by the International Futures (IFs) model v7.89 (Hughes et al., 2011).
Population and age structure data for China for 2015 and future years are obtained from the SSP
dataset (Samir and Lutz, 2017; Riahi et al., 2017), because China's current-policy scenario is
built upon the SSP2 scenario and the carbon-neutral scenario is built upon the SSP1 scenario
(Cheng et al., 2021b; Tong et al., 2020). The gridded population data for China on a $0.5° × 0.5°$
spatial resolution for 2015 and future years under each SSP scenario are developed by Huang et
al. (2019).
With the baseline mortality, the population data and the age-structure data, we calculate, grid
cell by grid cell, the age-specific baseline mortality rate under present and future scenarios. The
impacts of transboundary pollution on mortality in China is calculated as the difference between
mortality associated with $PM_{2.5}$ simulated by the full anthropogenic emissions for China and
foreign countries, and mortality associated with $PM_{2.5}$ simulated by excluding foreign emissions
from the full anthropogenic emissions.
**3. Results**
**3.1 Achievability of future air quality goals in China**.
In 2015, the national mean population-weighted $PM_{2.5}$ over China is about 48 $\mu g\ m^{-3}$ (Fig. 1)
after the observation-based correction, consistent with previous studies (48~55 $\mu g\ m^{-3}$) that used
various models (Burnett et al., 2018; Cheng et al., 2021a, 2021b; Tang et al., 2022; Zhang and
Cao, 2015). The correction reduces the overestimation in the model by roughly 16% (Fig. S2);
details about the correction approach are described in Method. From 2015 to 2030 and 2060,
there is a remarkable decreasing trend in China's annual mean population-weighted $PM_{2.5}$
concentrations under plausible futures (Fig. 1). In 2030, achieving the 35 $\mu g\ m^{-3}$ goal on a
national average level would be feasible even under the fossil fuel-intensive pathways in China
(current-policy) and foreign countries (SSP370), yet the most polluted provinces might not be
able to achieve the goal under such scenarios (upper whiskers in Fig. 1). In 2060, achieving the
WHO AQG goal of 5 $\mu g\ m^{-3}$ would be highly unlikely under any emission pathway analyzed in
this study (Fig. 1).
Considering air quality goals at the city level in China (35 $\mu g\ m^{-3}$ in 2015 and 2030; 5 $\mu g\ m^{-3}$ in
2060), the fraction of cities achieving air quality goals increases considerably as China and
foreign countries transition from the fossil fuel-intensive to the low-carbon pathway, yet the
achievement is not fully attainable in all cities. In 2015, only 30% of 365 cities in the national
$PM_{2.5}$ monitoring network has an annual mean population-weighted $PM_{2.5}$ below the 35 $\mu g\ m^{-3}$
threshold (Fig. 2a). In 2030, the percentage increases considerably (Fig. 2b). Under the current-
policy emission pathway in China (the current clean air policies and Nationally Determined
Contribution pledges), roughly 65% of Chinese cities (average of the three foreign scenarios) is
able to achieve the goal, doubling the percentage in 2015. Under the carbon-neutral emission
pathway in China (stringent clean air policies and carbon neutrality commitments), the
percentage further increases to about 92% (average of the three foreign scenarios). However,
even the cleanest emission pathway in both China and foreign countries cannot allow all cities to
attain the 35 $\mu g\ m^{-3}$ goal. In 2060 (Fig. 2c), the WHO AQG goal of 5 $\mu g\ m^{-3}$ is not achievable



for the majority of cities ($\geq$ 75%) in China. Emission pathways adopted by foreign countries can affect up to 6% of cities achieving the AQG goal, and the influence could be even larger over border regions (as will be shown in Fig. 3).

**3.2 Transboundary impacts on PM$_{2.5}$ concentrations in China**.

In 2015, transboundary pollution contributes about 3.8 µg m$^{-3}$ population-weighted PM$_{2.5}$ to China (Fig. 3a), accounting for roughly 8% of the total population-weighted PM$_{2.5}$ (Fig. 4). In the future, transboundary pollution becomes increasingly important in China, as the share of transboundary pollution in China's total population-weighted PM$_{2.5}$ increases to 12%~22% in 2060 (Fig. 4).

For future PM$_{2.5}$ concentrations, the contribution of transboundary pollution to PM$_{2.5}$ in China decreases as foreign countries and China undergo the low-carbon pathways (Fig. 3a). In 2030, under the current-policy scenario in China, transboundary contributions to PM$_{2.5}$ in China would be reduced by 1.2 µg m$^{-3}$ (29%) as foreign countries transition from the fossil fuel-intensive (SSP370) to the low-carbon (SSP119) scenario. By 2060, the difference would be increased to 1.8 µg m$^{-3}$ (45%). The transboundary pollution will also depend on Chinese domestic emissions, because of their chemical interactions with foreign-transported pollution (Xu et al., 2022). In 2030, under the SSP370 scenario in foreign countries, transboundary contributions to PM$_{2.5}$ in China could be reduced by 0.6 µg m$^{-3}$ (14%) as China transitions from the current-policy to the carbon-neutral scenario. In 2060, the PM$_{2.5}$ reduction could be increased to 1.8 µg m$^{-3}$ (45%).

The direct atmospheric transport and the chemical interactions play different roles in transboundary pollution over different regions in China. Over North China (outlined in Fig. 3c-o), the influence of chemical interactions on transboundary pollution is prominent, making China's domestic emission pathway an important driver to transboundary pollution and to the achievement of its air quality goals. In 2030, transboundary pollution contributes 3 to 10 µg m$^{-3}$ PM$_{2.5}$ concentrations even under China's carbon-neutral scenario (Fig. 3b and Fig. 3i-l), greatly increasing the difficulty for this region to achieve the 35 µg m$^{-3}$ goal. In 2060, if China adopts the current-policy pathway (Fig. 3f-h), achieving the WHO AQG goal of 5 µg m$^{-3}$ would be highly unlikely for the majority of North China, as transboundary pollution alone would contribute roughly 3~10 µg m$^{-3}$ of PM$_{2.5}$ concentration (Fig. 3b and Fig. 3f-h). Alternatively, adopting the carbon-neutral rather than the current-policy pathway in China reduces roughly 41%~58% of transboundary PM$_{2.5}$ over North China in 2060 (Fig. 3b), making it possible for the region to achieve the 5 µg m$^{-3}$ goal. Thus, a low emission pathway for China has a large co-benefit on reducing transboundary pollution exerted upon its populous northern area by reducing the aforementioned chemical interactions.

The western border provinces of Yunnan and Xinjiang (denoted in Fig. 3c-o) are also influenced substantially by transboundary pollution, with the magnitude of transboundary pollution determined predominantly by foreign (but not Chinese) emission pathways. For example, in 2060, under China's carbon-neutral emission pathway, transboundary PM$_{2.5}$ over Yunnan decreases from 6 µg m$^{-3}$ to 3 µg m$^{-3}$ (a 50% reduction) as the foreign pathway switches from fossil fuel-intensive (SSP370; Fig. 3m) to low-carbon (SSP119; Fig. 3o), reflecting the considerable impact of Southern Asian pollution to China (Jiang et al., 2013). Over Xinjiang, the



transboundary pollution driven predominantly by direct atmospheric transport can reach 6 µg m$^{-3}$
in many scenarios (i.e., current-policy plus SSP370 for 2030 and 2060; and carbon-neutral plus
SSP245 for 2030), indicating the important influence of anthropogenic emissions from Central
Asia in the future that has hardly been investigated in previous studies.
**3.3 Health threats by transboundary pollution in China.**
Figure 5 shows our estimated PM$_{2.5}$-associated premature deaths in 2015 and the future. Our
estimated PM$_{2.5}$-associated premature deaths in China in 2015 (2.03 million) is comparable with
other studies (2 to 2.4 million; Burnett et al., 2018; Geng et al., 2021; Tang et al., 2022). Our
estimated increasing trend of premature deaths in China from SSP370 to SSP119 is also
consistent in previous works (Tang et al., 2022; Yang et al., 2022), which is driven primarily by
population ageing (Fig. S3). Our estimated premature deaths in 2030 and 2060 under the current-
policy emission scenario (2.68 to 2.82 million for 2030; 1.8 to 2.05 million for 2060) is
substantially lower than those in Tang et al. (2022) (3.5 to 4 million for 2030; 6.5 to 7.5 million
for 2060). The difference is primarily because Tang et al. (2022) fixed the baseline mortality
rates in future years at the 2015 level, while our future baseline mortality rates from the IFs are
projected on the basis of income, education and technology advancement, and other factors
(Hughes et al., 2011).
The PM$_{2.5}$ concentrations contributed by transboundary pollution can lead to an extra number
of people (i.e., excess population) exposed to PM$_{2.5}$ concentrations above the targeted air quality
levels in China (35 µg m$^{-3}$ in 2015 and 2030; and 5 µg m$^{-3}$ in 2060), which may lead to potential
health threats. In the future, the number of excess population due to transboundary pollution
depends on both foreign and China's emission pathways (Fig. 6a). In 2030, with Chinese
emissions following the carbon-neutral scenario, adopting the low-carbon (SSP119) rather than
the fossil fuel-intensive pathway (SSP370) in foreign countries could avoid 10 million Chinese
people from being exposed to PM$_{2.5}$ concentrations above 35 µg m$^{-3}$. In 2060, 5 million people
could be avoided from being exposed to PM$_{2.5}$ concentrations above 5 µg m$^{-3}$ if the foreign
scenario switches from SSP370 to SSP119. These results indicate remarkable health benefits that
the low-carbon emission pathway in foreign countries would bring to China.
For a given future year and foreign emission scenario, the excess population due to
transboundary pollution tends to be larger when China adopts the carbon-neutral pathway than
when China adopts the current-policy pathway (Fig. 6a). This reflects the increasing influence of
transboundary pollution on air quality in China as China's overall PM$_{2.5}$ concentrations drop
sharply towards the air quality goals under the carbon-neutral pathway (Fig. 1).
Another measure of health threat that transboundary pollution could exert upon China is PM$_{2.5}$-
related premature deaths. As shown in Fig. 6b, there is an obvious decreasing trend of
transboundary-contributed PM$_{2.5}$-related premature deaths in China as foreign countries and
China transition from the fossil fuel-intensive to the low-carbon pathway. In 2030, adopting the
low-carbon pathway (SSP119) in foreign countries would avoid 41% (178,000 under China's
carbon-neutral pathway) to 45% (207,000 under China's current-policy pathway) of premature
deaths that would occur under the fossil fuel-intensive pathway (SSP370) in foreign countries. In
2060, the avoidance would be as large as 76% (270,000 under China's current-policy pathway)



to 91% (63,000 under China's carbon-neutral pathway). In addition, China's low carbon emission pathway could also bring considerable health benefits through reducing the chemical interaction-related transboundary pollution and associated premature deaths in China. In 2030, adopting the carbon-neutral pathway in China would avoid 99,000 (SSP245) to 211,000 (SSP119) people from transboundary pollution-associated premature deaths relative to adopting the current-policy emission pathway. In 2060, the avoided deaths would be 76,000 (SSP119) to 283,000 (SSP370). These findings highlight the considerable health benefits to China if foreign countries and China could adopt the low-carbon emission pathways coincidently.

## 4. Discussion

This study reveals the increasingly important role that transboundary pollution would play in the achievement of future air quality goals and the protection of public health in China. The magnitude of transboundary pollution depends on both Chinese and foreign emissions, given the direct pollution transport and the indirect impact through chemical interactions between transported and China's locally emitted pollutants. Adopting the low-carbon (SSP119) instead of the fossil fuel-intensive (SSP370) pathway in foreign countries would avoid millions of Chinese people (Fig. 6a) from being exposed to $PM_{2.5}$ concentrations above the targeted air quality levels in 2030 (35 µg m$^{-3}$) and 2060 (5 µg m$^{-3}$), and would avoid 63,000~270,000 of transboundary $PM_{2.5}$-associated mortality in China in 2060 (Fig. 6b). Adopting the carbon-neutral instead of current-policy pathway in China would avoid 76,000~283,000 premature mortality associated with transboundary pollution in 2060 (Fig. 6b). If China and foreign countries undergo the low-carbon pathways coincidently, transboundary pollution in China would be reduced by 63% relative to adopting a fossil fuel-intensive emission pathway in both regions (Fig. 4a), and could avoid 386,000 premature deaths in China (Fig. 6b). Cutting foreign emissions are particularly effective at reducing transboundary pollution upon the western border provinces of Yunnan and Xinjiang that are dominated by direct transport. Fully achieving the WHO AQG goal of 5 µg m$^{-3}$ over the populous North China would be possible only when both China and foreign counties adopt the low-carbon pathways (carbon-neutral and SSP119, respectively).

The importance of transboundary pollution is not confined in China. In the future, significant emission changes are expected in many developing countries, affecting air quality locally and in the downwind regions. These developing countries are often financially and/or technologically less capable to control emissions by themselves. Thus, enhanced external aids would be essential for these developing countries to undergo a low-carbon development in the future, which in turn would benefit air quality and public health of the entire globe. Such aids could be deployed through global or inter-regional cooperation programs such as the Paris Agreement and the Belt and Road Initiative. Promoting the environmental cooperation is particularly meaningful nowadays when the 2019 coronavirus (COVID-19) pandemic, the Russia-Ukraine conflict and the emergence of regional rivalry (e.g., the Sino-US trade war; Du et al., 2020) disrupt the global society and environment, threatening cooperation at all levels.

Uncertainties arise from several factors in this study. The future development pathway of a country is highly uncertain, leading to a wide spread of projected emission trajectories in the future. We thereby use a set of emission projections to represent the plausible range of future emissions. The simulation of $PM_{2.5}$ is subject to uncertainties in aerosol chemical and physical



processes, such as the wet deposition of nitrate (Luo et al., 2020) and the simplified secondary
organic aerosol formation scheme (Pai et al., 2020). Our correction to the simulated $PM_{2.5}$
concentrations using ground-based observations could reduce the uncertainty to a certain degree.
Future population and age structure change are projected based on their historical relationships
with GDP and urbanization (O'Neill et al., 2020; Riahi et al., 2017). Thus, they may introduce
biases if the future development of global GDP and urbanization deviates from the historical path
(e.g., due to the emergence of anti-globalization (Dür et al., 2020) and regional rivalry (O'Neill
et al., 2014; van Vuuren et al., 2014). There are additional uncertainties from $PM_{2.5}$-related death
estimates due to the limited epidemiology evidence and statistical estimation of the GEMM
model, such as the influences of particulate species and size on health outcomes (Burnett et al.,
2018). Besides, we do not consider potential influences of climate change and the change of
natural emissions on $PM_{2.5}$ and transboundary pollution, yet their influences are found small
compared to the influence of anthropogenic emissions (Hong et al., 2019; Jiang et al., 2013;
Silva et al., 2017).
**Data availability**
The global SSP-RCP emissions data for 2015 and future scenarios, and the area-weighted
$PM_{2.5}$ concentrations in China for 2015 and future scenarios are available upon request to the
corresponding author. All other data used in this study are publicly available and can be
downloaded from the following links. (1) China's future emission scenarios 2015–
2060: http://www.meicmodel.org/dataset-dpec.html. (2) Chinese future population data:
https://www.scidb.cn/en/detail?dataSetId=73c1ddbd79e54638bd0ca2a6bd48e3ff&dataSetType=
organization. (3) 2015 baseline mortality rate: https://gbd2017.healthdata.org/gbd-search/. (4)
Future baseline mortality rate projection: https://www.ifs.du.edu/IFs/frm_MortCohorts/.
**Code availability**
The GEOS-Chem model v13.2.1 source code used for $PM_{2.5}$ concentration simulations is
available at: http://www.geos-chem.org. The SSP-RCP emission harmonization source code is
available at http://software.ene.iiasa.ac.at/aneris/. All computer codes generated during this study
are available from the corresponding authors upon reasonable request.
**Acknowledgements**

This work was supported by the National Natural Science Foundation of China (42075175), the
China Postdoctoral Science Foundation (2021M700191) and the Peking University Boya
Postdoctoral Fellowship.
**Author contributions**
J.L. led the study. J.X. and J.L. designed the study. J.X. performed the model simulations and
conducted the data analysis. D.T. provided China's future emissions data. L.C. provided SSP-
RCP emission harmonization and health impact assessment methods. J.X. wrote the manuscript
with inputs from J.L. All authors commented on the manuscript.

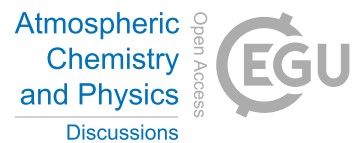
**Competing interests**

The authors declare that they have no conflict of interest.

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



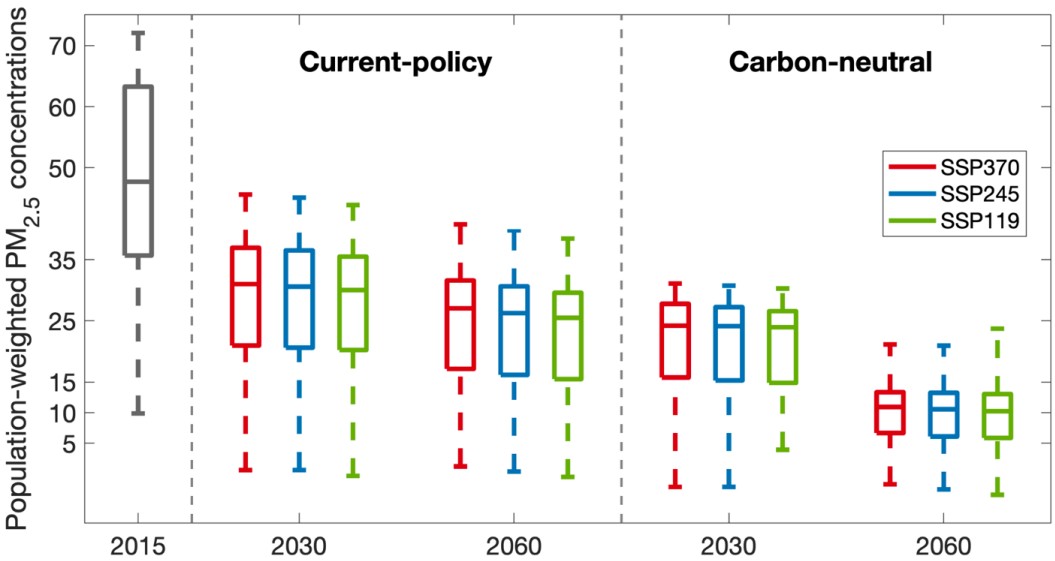

**Figure 1.** Population-weighted PM$_{2.5}$ concentrations over China. Box-and-whisker plots represent 5th, 25th, 75th, and 95th percentiles of provincial population-weighted PM$_{2.5}$ concentrations in 2015, 2030 and 2060. Lines in the middle of each box represent the national mean population-weighted PM$_{2.5}$ concentrations. Future emission scenarios in China are labeled as text at the top and in foreign countries are represented by colors according to the legend.

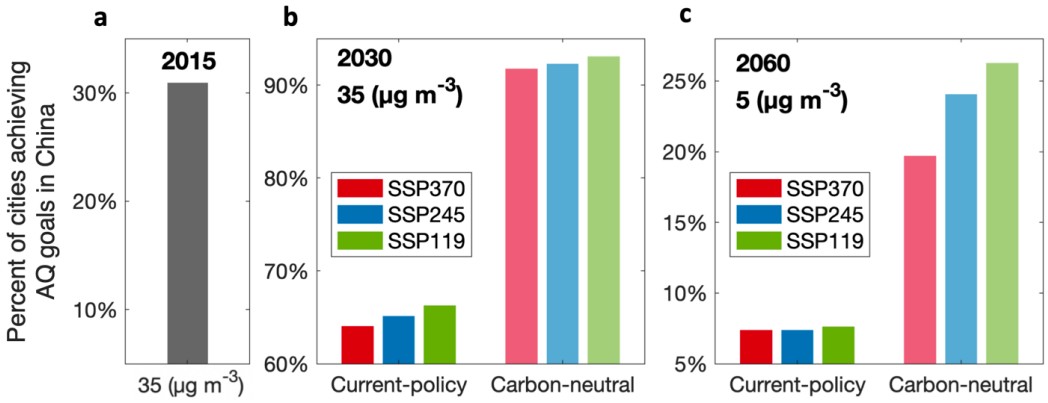

**Figure 2.** Percent of cities achieving air quality goals in China. **(a)** Percent of cities in China with an annual mean population-weighted PM$_{2.5}$ concentration below 35 µg m$^{-3}$ in 2015. **(b-c)** Percent of cities with an annual mean population-weighted PM$_{2.5}$ concentration achieving the 35 µg m$^{-3}$ goal in 2030 (b) and the 5 µg m$^{-3}$ goal in 2060 (c). PM$_{2.5}$ concentrations are simulated under different future emission scenarios in China (current-policy and carbon-neutral) and foreign countries (represented by colors following the legend).

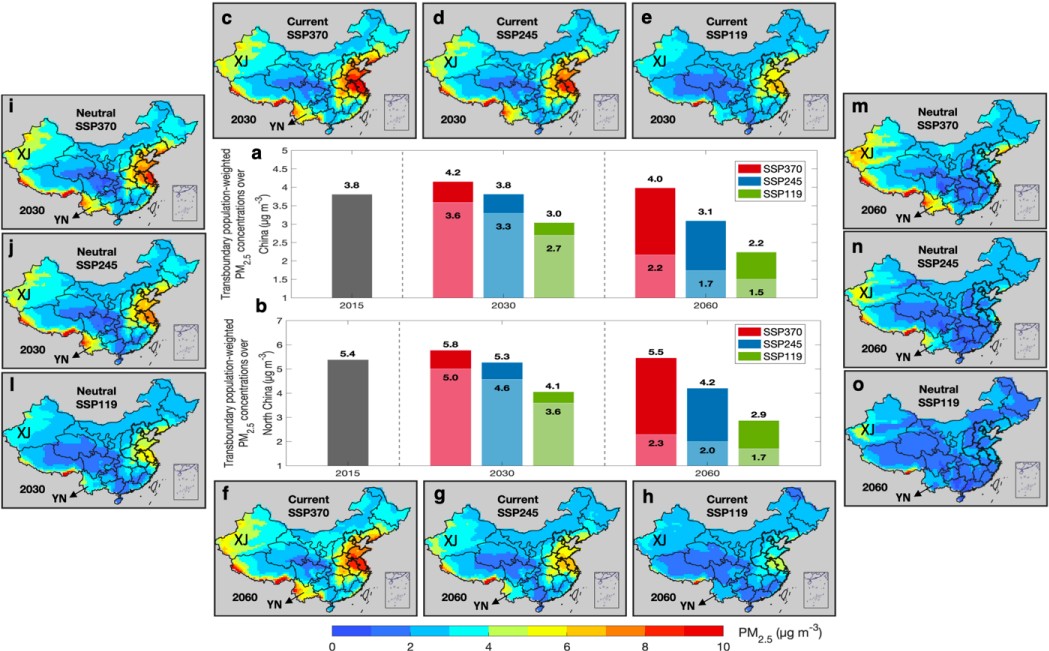

**Figure 3.** Contributions of transboundary pollution to PM$_{2.5}$ concentrations over China. **(a)** Transboundary contributions to national annual mean population-weighted PM$_{2.5}$ in China in 2015, 2030 and 2060. Future scenarios are estimated by different emission scenarios in China represented by light (carbon-neutral scenario) and dark shadings (current-policy scenario), along with different emission scenarios in other countries (SSP370, SSP245, SSP119) represented by colors according to the legend. Text on top of each bar represents the transboundary-contributed population-weighted PM$_{2.5}$ under China's current-policy emission scenarios. Text in the light shading of each bar represents the transboundary-contributed population-weighted PM$_{2.5}$ under China's carbon-neutral emission scenarios. **(b)** Same as a, but for North China. **(c-o)** Spatial distributions of transboundary-contributed annual mean PM$_{2.5}$ concentrations over China in 2030 and 2060 under different emission scenarios in China and in other countries. YN represents Yunnan province. XJ represents Xinjiang province.

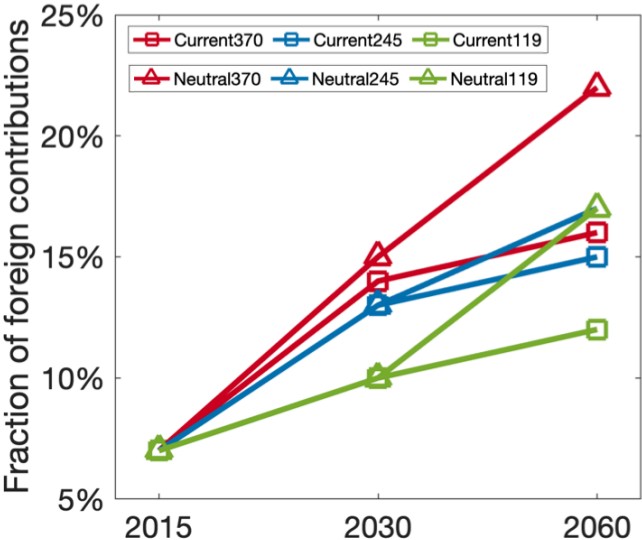

**Figure 4.** Fractional transboundary contributions to $PM_{2.5}$ concentrations over China. The percentage fraction of transboundary-contributed $PM_{2.5}$ in China's total population-weighted $PM_{2.5}$ in 2015, 2030 and 2060. Future anthropogenic emission scenarios are represented by different colors and markers following the legend.

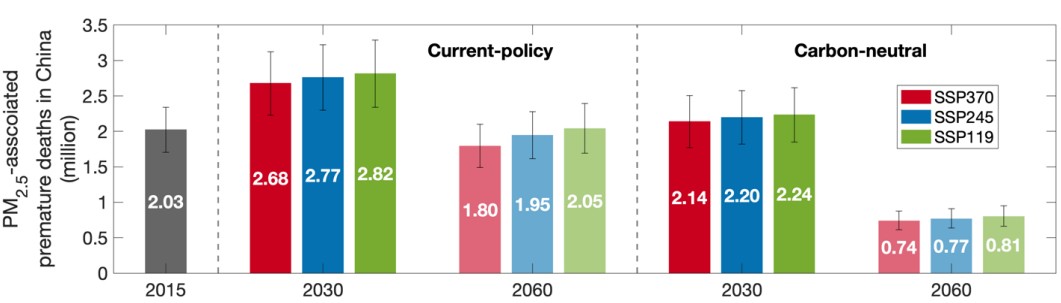

**Figure 5.** $PM_{2.5}$-associated premature deaths in China. Total $PM_{2.5}$-asscoiated premature deaths in China for 2015, 2030 and 2060 under each emission scenario in China (denoted as text at the top) and in other countries (represented by colors in the legend). Numbers denote the estimated deaths in each scenario. Error bars represent the 95% confidence interval of the RR function.





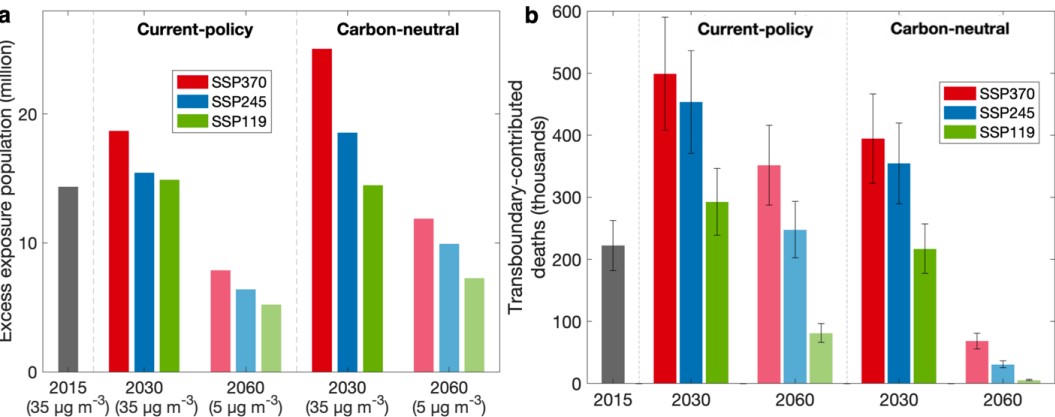

**Figure 6.** Potential health threats associated with transboundary pollution in China. **(a)** Population exposed to an annual mean population-weighted $PM_{2.5}$ concentration above the goals (35 µg m$^{-3}$ for 2015 and 2030; 5 µg m$^{-3}$ for 2060) due to transboundary contributions of $PM_{2.5}$ in China under different emission scenarios in China (denoted as text at the top) and in other countries (represented by colors in the legend). **(b)** Transboundary-contributed $PM_{2.5}$-related premature mortality in China under different emission scenarios in China (denoted as text at the top) and in other countries (represented by colors in the legend). Error bars represent the 95% confidence interval of the RR function.



**Table 1.** Description of future scenario settings

| Scenarios | Socioeconomic pathway | Climate target | Pollution control strength | PM$_{2.5}$ emission level | Key features |
|---|---|---|---|---|---|
| Foreign | | | | | |
| SSP119 | Sustainability | 1.9 W m$^{-2}$ (1.5 ℃) | Strong | Low | Strong economic growth via sustainable pathway. Incomes increase substantially and inequality within and between countries is greatly decreased. Significantly lower demand for energy- and resource-intensive agricultural commodities. Effective pollution controls result in substantial reductions in air pollutant emissions. |
| SSP245 | Middle-of-the-road | 4.5 W m$^{-2}$ (3 ℃) | Medium | Medium | An intermediate case between SSP1 and SSP3. |
| SSP370 | Regional rivalry | 7.0 W m$^{-2}$ (~4 ℃) | Weak | High | High inequity between countries. GDP growth is low and concentrated in current high-income nations, while population growth is focused in low- and middle- income countries. Energy system is coal-intensive. The implementation of pollution controls is delayed and less effective. |
| China | | | | | |
| Current-policy | Middle-of-the-road (SSP2) | 4.5 W m$^{-2}$ (3 ℃) | Medium | Medium | Achieve China's Nationally Determined Contribution (NDC) pledges and the national PM$_{2.5}$ air quality standard (i.e. 35 μg m$^{-3}$) by 2030, elucidating China's future air pollution mitigation pathway towards all the released and determined upcoming clean air policies since 2015. |
| Carbon-neutral | Sustainability (SSP1) | Net-zero CO$_2$ emissions in 2060 | Strong | Low | Pursue China's carbon-neutral commitment and the WHO's old PM$_{2.5}$ guideline (10 μg m$^{-3}$) by 2060. It implements the best available end-of-pipe technologies and more stringent pollution control policies than the current-policy. |




1    **Table 2.** Simulation configurations

| Simulation type | Anthropogenic Emissions | | Emission year | Met field year |
|---|---|---|---|---|
| | China | Foreign countries | | |
| Base_2015 | MEIC | CEDS | 2015 | 2015 |
| China_2015 | MEIC | None | | |
| Base_current_SSP119_2030 | Current-policy | SSP119 | 2030 | |
| Base_current_SSP245_2030 | Current-policy | SSP245 | | |
| Base_current_SSP370_2030 | Current-policy | SSP370 | | |
| China_current_2030 | Current-policy | None | | |
| Base_neutral_SSP119_2030 | Carbon-neutral | SSP119 | | |
| Base_neutral_SSP245_2030 | Carbon-neutral | SSP245 | | |
| Base_neutral_SSP370_2030 | Carbon-neutral | SSP370 | | |
| China_neutral_2030 | Carbon-neutral | None | | |
| Base_current_SSP119_2060 | Current-policy | SSP119 | 2060 | |
| Base_current_SSP245_2060 | Current-policy | SSP245 | | |
| Base_current_SSP370_2060 | Current-policy | SSP370 | | |
| China_current_2060 | Current-policy | None | | |
| Base_neutral_SSP119_2060 | Carbon-neutral | SSP119 | | |
| Base_neutral_SSP245_2060 | Carbon-neutral | SSP245 | | |
| Base_neutral_SSP370_2060 | Carbon-neutral | SSP370 | | |
| China_neutral_2060 | Carbon-neutral | None | | |

