# Peer review of "The underappreciated role of transboundary pollution in future air quality and health improvements in China"

_Atmospheric Chemistry and Physics, 2023_

## Author Response (AR1)

**Authors' Response to Comments from Referee #1**

The authors assess the extent to which future changes in foreign transboundary pollution would influence the achievability of air quality goals in 2030 and 2060 for China in this study. The results are interesting and valuable and should support decision making activities for regional environmental policy. Overall, the manuscript will be suitable for publication following minor revisions to address the points listed below.

**Response: We sincerely thank the Referee #1 for taking the time to review our paper and for providing constructive suggestions for improvement. Reponses to these comments are provided below.**

My major concerns are as follows:

I appreciate the effort of the authors in collecting air quality and health data, designing emissions scenarios, and adjusting their models to do the calculations. The methods section though could perhaps be shortened significantly, as it seems the authors are using the same model setup as their recently published work (Xu et al, 2023). In fact, they copy several paragraphs, word-for-word, from Xu et al. (2023) [compare text starting page 5, line 26, to page 6, lines 32, of the present manuscript to sections 2.1 and 2.2 of their previous paper]. While copying from their own paper, this is still technically plagiarism. I've seen papers summarily rejected from ACPD for far less. If the needed text is already published, it should simply be referenced. If there are differences that warrant restatement, then it should be presented in original fashion. They should examine other sections of their methods description for similar situations, removing "boilerplate" descriptions where possible.

**Response: Thanks for point this out. We removed the common parts and only restated important configuration settings in original fashion on page 5-6 to the following:**

**"We use the GEOS-Chem model to simulate PM$_{2.5}$ concentrations in China and other Asian countries under each emission scenario and year. Detailed descriptions of the model configurations can be found in Xu et al. (2023). Briefly, we use the Flex-Grid capability of the GEOS-Chem classic model v13.2.1 to simulate PM$_{2.5}$ concentrations over Asia and its adjacent regions (11° S–60° N, 30°–150° E; covering the entire Asia, eastern Africa and eastern Europe). We use MERRA-2 assimilated meteorological data provided by the Global Modeling and Assimilation Office (GMAO) at NASA Goddard Space Flight Center to drive the model. Our simulation is conducted at a horizontal resolution of 0.5° latitude × 0.625° longitude with 47 vertical levels between the surface and ∼ 0.01 hPa. Chemical boundary conditions are taken from corresponding global simulations under each emission scenario in Table 2 at a resolution of 2° latitude × 2.5° longitude. The regeneration of boundary conditions under each emission scenario could ensure the inclusion of pollution transported from countries both within and**

**outside the Flex-Grid domain as transboundary pollution to China. We spin up every simulation for 1 month to remove the effects of initial conditions.**

**Anthropogenic emissions for the base year (2015) for China are taken from the Multi-resolution Emission Inventory (MEIC) for 2015 (Zheng et al., 2018), and for the rest of the world are taken from the Community Emissions Data System (CEDS) version 2 for 2015 (https://data.pnnl.gov/dataset/CEDS-4-21-21). For future simulations, anthropogenic emissions for China and foreign countries for each scenario are described above and are specified in Table 2. Other emissions are default in GEOS-Chem following Xu et al. (2023) and are fixed in all present and future scenarios."**

The authors provide valuable information on how different domestic and foreign environmental and climate strategies would affect future transboundary pollution and public health in China. Considering their nested domain, I'm curious about what the so-called "foreign emission" includes in their analyses. Do they only focus on the emissions of Asian countries included in their nested domain or the impacts of the emissions from America and Europe could also be accounted? Have the boundary conditions been regenerated for each scenario?

**Response: Thanks for pointing this out. Yes, we regenerated boundary conditions from global 2°x2.5° simulations for each scenario, so foreign emissions include emissions from the entire globe except China. We described the boundary conditions in more detail on page 6 as the following "Chemical boundary conditions are taken from corresponding global simulations under each emission scenario in Table 2 at a resolution of 2° latitude × 2.5° longitude. The regeneration of boundary conditions under each emission scenario could ensure the inclusion of pollution transported from countries both within and outside the Flex-Grid domain as transboundary pollution to China."**

In addition to the general impacts of the "foreign emission changes," readers might also want to know where the foreign impacts could be from, for example as least within and outside the nested domain. As suggested by the authors, emissions are continuously reduced in some countries while less regulated in others. The region-specific analyses would tell what kind of international cooperation would be urgently needed to further improve future air quality.

**Response: Thanks for the suggestion. We conducted additional simulations to separate transboundary pollution from countries outside and within the Flex-Grid domain as Fig. S5. We added the description of the simulations to page 7:**

**"We also conduct sensitivity simulations at a resolution of 2° x 2.5° to separate transboundary contributions to China's future PM$_{2.5}$ concentrations from anthropogenic emissions within or outside the Flex-Grid domain of the model (11° S–60° N, 30°–150° E, including the entire Asia, eastern Africa and eastern Europe) under the current-policy and SSP370 scenario. Contributions of foreign anthropogenic emissions from within the Flex-Grid domain are calculated as the difference of a simulation without foreign anthropogenic emissions outside the Flex-Grid domain and a simulation without foreign anthropogenic**

emissions over the globe. Contributions of foreign anthropogenic emissions from outside the Flex-Grid domain are calculated as the difference between the total foreign contributions and foreign contributions from Flex-Grid countries."

We also added the discussion on the contributions of foreign anthropogenic emissions from within or outside the Flex-Grid domain on page 10:

"Transboundary pollution to China could arise from emissions of countries within and outside the Flex-Grid domain of the model (11° S–60° N, 30°–150° E, including the entire Asia, eastern Africa and eastern Europe). Thus, we conduct a sensitivity simulation driven by China's current-policy and foreign countries' SSP370 emissions in 2030 and 2060 as an example to explore the relative importance of anthropogenic emissions from countries within and outside the Flex-Grid domain for transboundary pollution to China. Fig. S5 shows that anthropogenic emissions from countries within the Flex-Grid domain dominate the transboundary pollution to China in all seasons (>80%), and their contributions remain in 2030 and 2060. Contributions of anthropogenic emissions from countries outside the Flex-Grid domain is larger in April and October when westerly winds prevail and wet deposition is not as strong as in July (Jiang et al., 2013; Leibensperger et al., 2011; Ni et al., 2018). North China is more influenced by emissions outside the Flex-Grid domain (13% in April and 24% in October) than China on average is (7% in April and 13% in October), indicating that North China is more influenced by emissions outside Asia, such as Europe and North America. Leibensperger et al. (2011) have also found that $NO_x$ emission in the US could lead to an enhancement of $PM_{2.5}$ over North China.

The authors suggest that the uncertainty of this study can arise from several factors. It would be helpful to provide more quantitative discussions on the uncertainty. At least those induced by the health assessment can be easily quantified.

**Response: We estimated the uncertainty of $PM_{2.5}$-related premature mortality as the lower and upper bounds of a 95% confidence interval around mean attributable mortality in each scenario in Fig. 5 and Fig. 6b. We also added quantitative discussions on the uncertainty of health assessment as the following in the Discussion section on page 13: "We estimate the overall uncertainties of $PM_{2.5}$-related death in each scenario as 95% CI in the main text"**

Specific comments:

Page 1 L19-25: I understand that the authors would like to present their general findings here. However, the results presented here are easy to imagine without any numerical experiment being conducted, since more strict emission control strategies would result in larger air quality and health benefits for sure. I suggest more quantitative results could be presented in the abstract according to the sensitivity experiments conducted in this study.

**Response: Thanks for the suggestion. We included quantitative results in the abstract as the following:**

"We find that in 2030, under the current-policy scenario in China, transboundary contributions to population-weighted PM$_{2.5}$ in China would be reduced by 29% (1.2 μg m$^{-3}$) as foreign countries transition from the fossil fuel-intensive to the low-carbon scenario. By 2060, the difference would be increased to 45% (1.8 μg m$^{-3}$). Adopting the low-carbon instead of the fossil fuel-intensive pathway in foreign countries would avoid 10 million Chinese people from being exposed to PM$_{2.5}$ concentrations above China's Ambient Air Quality Standard (35 μg m$^{-3}$) in 2030 and 5 million Chinese people from being exposed to PM$_{2.5}$ concentrations above the World Health Organization Air Quality Guideline (5 μg m$^{-3}$) in 2060."

Page 3 L7-11: Can any detailed example be provided in terms of the interactions between the impacts of foreign emissions and the local chemistry?

Response: We added the detailed mechanism of the chemical interactions on page 3 "Xu et al. (2023) found that the transport and transformation of non-methane volatile organic compounds (NMVOCs) from foreign countries could enhance the atmospheric oxidizing capacity and facilitated the oxidation of Chinese nitrogen oxides (NO$_x$) to form nitric acid (HNO$_3$) and nitrate over North China (referred to as eastern China in that study), leading to the considerable foreign contributions to China's air pollution".

Pager 3 L32-35: Can any quantitative description, instead of "smaller," be provided here so that the reader could have sense of to what extent the meteorology could affect the PM2.5 concentrations in the future?

Response: Thanks for pointing this out. We revised the description on page 3 to the following: "We do not consider the effects of physical climate change (e.g., temperature and precipitation) on future transboundary pollution because Liu et al. (2021) found that future PM$_{2.5}$ changes in China driven by anthropogenic emission reduction was 7 times more than the changes due to meteorological fields by 2050 in both SSP126 and SSP585 scenarios. Other studies have also indicted that the impact of climate change on future PM$_{2.5}$ were smaller than those of anthropogenic emissions of pollutants (Hong et al., 2019; Jiang et al., 2013; Silva et al., 2017)."

Page 4 L10: It would be good to add a reference here for the bias correction method.

Response: We added the reference Xu et al. (2023).

Page 4 L20-24: SSP1-5 might not necessarily need to be explained scenario by scenario, since they are not used in this study.

Response: We removed the description of SSP4 and SSP5 scenarios as they were not used in this study, while keeping the other three scenarios on page 4 as the following:

"We select three scenarios to represent low, medium and high emission cases: SSP1 - sustainability, SSP2 - middle-of-the-road, SSP3 - regional rivalry."

Page 4 L43: What are the 2019 emissions used for?

**Response: 2019 emissions from the CEDS inventory were used to harmonize future projections of SSP-RCP emissions. We added a detailed description on page 5 as the following:**

**"The original SSP-RCP anthropogenic emissions future projection starts from 2015. Here, we update the start year to 2019, which is the latest year that anthropogenic emissions from the Community Emissions Data System (CEDS; Hoesly et al., 2018) are available. Because CEDS historical emissions and SSP-RCP future emissions were developed upon different assumptions and methods, it is important to harmonize the two datasets to ensure smooth transitions between the two sets of emissions trajectories (Gidden et al., 2019). The update of the harmonization year from 2015 to 2019 in this study could better represent the actual emission trajectory between 2015 and 2019. We also use the most recently developed CEDS emissions version 2 (https://data.pnnl.gov/dataset/CEDS-4-21-21) to harmonize the future SSP-RCP emission projections as the new emissions can better represent the historical trend of pollutant emissions …"**

Page 5 L34: Could the impacts from countries other than those included in the nested domain be considered in this study?

**Response: Yes. We added Fig. S5 to separate the contributions of anthropogenic emissions from countries within and outside the Flex-Grid domain. Detailed revisions have been elaborated above in the referee's "major concern" section.**

Page 5 L41-42: Are the boundary conditions updated in each sensitivity experiment?

**Response: Yes. We included the description to page 6 as the following: "Chemical boundary conditions are taken from corresponding global simulations under each emission scenario in Table 2 at a resolution of 2° latitude × 2.5° longitude. The regeneration of boundary conditions under each emission scenario could ensure the inclusion of pollution transported from countries both within and outside the Flex-Grid domain as transboundary pollution to China."**

Page 7 L6-8: I assume the ground-based PM2.5 concentrations are integrated results of both anthropogenic and natural emissions. It would be helpful to explain why the bias correction is only conducted in anthropogenic pollution-dominated grids.

**Response: We explained the reason on page 6 as the following and added Fig. S3 to illustrate the point:**

**"We correct simulated $PM_{2.5}$ concentrations in anthropogenic concentration-dominated grid cells (where anthropogenic emissions exceed natural emissions) by observations located in anthropogenic concentration-dominated grid cells. Similarly, we correct simulated**

concentrations in natural concentration-dominated grid cells (where natural emissions exceed anthropogenic emissions) by observations located in natural concentration-dominated grid cells. Natural and anthropogenic concentration-dominated grids in the model are shown in Fig. S3."

Page 8 L3-L6: Any causality here?

Response: We assumed that the cause-and-effect relationship between $PM_{2.5}$ exposure and mortality in the present remained in the future. Thus, we used the GEMM model that constructed $PM_{2.5}$-mortality hazard ratio function based on cohort studies of outdoor air pollution that covered the global exposure range in 2015. This model has been widely used in studying future $PM_{2.5}$-atrributed mortality. We revised the main text on page 8 to illustrate this point:

"Future cause-and-effect relationship between $PM_{2.5}$ exposure and mortality follows the $PM_{2.5}$-mortality hazard ratio function in the GEMM model, which has been widely used in the calculation of $PM_{2.5}$-associated mortality in the future (Hong et al., 2019; Liu et al., 2021; Yang et al., 2022)."

Page 9 L35: I'd be curious about where these transboundary PM2.5 are from. In there previous work (Xu et al., 2023), they included some information on sector contributions from transboundary sources.

Response: We conducted additional simulations on sectoral contributions to transboundary $PM_{2.5}$ and included the result as Figure S4. We included the simulation description to the method part on page 7 as the following:

"We calculate sectoral contributions of foreign anthropogenic emissions to China's $PM_{2.5}$ concentrations under China's current-policy and foreign countries SSP370 scenario at a resolution of 2° x 2.5° for January in 2030 and 2060. Sectoral contributions are calculated by taking the difference of a simulation that includes one sector of SSP370 foreign anthropogenic emissions (agriculture, industry, energy, traffic, residential combustion, solvent use, waste burning) at a time and a simulation without foreign anthropogenic emissions ("China_current_2030" and "China_current_2060" runs in Table 2)."

We also added the discussion on the result of sectoral contributions to page 10 as the following:

"As transboundary contributions are the largest under China's current-policy and foreign countries' SSP370 emission pathways (Fig. 3a), we further investigate major emission sectors in foreign countries emissions that contribute to China's future $PM_{2.5}$ concentrations. We take January as an example because Xu et al. (2023) found that transboundary contributions to China $PM_{2.5}$ were the largest in January. Fig. S4 reveals that agriculture, energy, industry, transportation and residential combustion emissions in foreign countries are major sources of

transboundary pollution to China's national average future PM$_{2.5}$ concentrations in 2030 and 2060 January, and that these sectors contribute roughly evenly (15%~23% for each source). There is pronounced spatial heterogeneity in the source attribution. Solvent use emissions in foreign countries make roughly 3 times larger contributions to North China (18% in 2030 and 2060) than to the national average (6%-7% in 2030 and 2060). This is in line with the result of Xu et al. (2023) that the NMVOCs are the primary drivers of transboundary PM$_{2.5}$ over North China. Residential combustion in foreign countries contributes about 16% in 2030 and 10% 2060 more PM$_{2.5}$ to Yunnan province than to the entire China, as Yunnan is mostly affected by emissions from South Asia (Jiang et al., 2013) where residential combustion is intensive (McDuffie et al., 2020)."

Page 11 L31-38: In practice, it would be difficult to take the external factors into consideration when designing domestic air quality or climate strategies. The most strict emission control measures should be considered as if no controls were conducted in surrounding countries. I would discuss what China would do under such kind of conditions rather than the benefit dependent on the uncertain external aids.

**Response: Thanks for the suggestions. We revised the description to the following on page 12:**

  **"External aids could be done through global cooperation programs such as the Paris Agreement or through inter-regional collaboration such as the Belt and Road Initiative."**

Page 11 L39-41: I would delete this part to avoid political comments in a scientific study.

**Response: Done.**

Page 12 L3: It would be good to provide what degree instead of "a certain degree."

**Response: Thanks for the suggestion. We wrote out the "a certain degree" in numbers as the following on page 13: "Our correction to the simulated PM$_{2.5}$ concentrations using ground-based observations could reduce the uncertainty by 15% to 18% (Fig. S2)."**

Pager 12 L12: I've been curious about how small the influence of meteorology is during the whole reading process. It would be good to explain more either here or in the introduction according to the cited studies.

**Response: We revised in the introduction on page 3 to quantify the effect of meteorology on future PM$_{2.5}$ concentrations as the following: "We do not consider the effects of physical climate change (e.g., temperature and precipitation) on future transboundary pollution because Liu et al. (2021) found that future PM$_{2.5}$ changes in China driven by anthropogenic emission reduction was 7 times more than the changes due to meteorological fields by 2050 in both SSP126 and SSP 585 scenarios. Other studies have also indicted that the impact of**

**climate change on future PM$_{2.5}$ were smaller than those of anthropogenic emissions of pollutants (Hong et al., 2019; Jiang et al., 2013; Silva et al., 2017)."**

Page 20: Units are missing in Fig. 1.

**Response: Added.**

**Authors' Response to Comments from Referee #2**

This study conduct GEOS-Chem chemical transport model simulations to investigate the influence of transboundary pollution on the air quality and human health under various future scenarios. The authors found that the adoption of the low-carbon, as opposed to the fossil fuel-intensive, pathway in foreign countries could significantly improve the air quality and therefore reduce the premature deaths in China. In addition, they found that should China adopt the carbon-neutral, as opposed to the current-policy, pathway, transboundary PM2.5 would also be reduced from the chemical interactions between foreign-transported and locally-emitted pollutants.

The topic is interesting and important, the analysis is comprehensive, figures are nicely presented and organized, and the manuscript is well written. I only have a few relatively minor suggestions for the authors to consider.

**Response: We thank the Referee #2 for providing constructive suggestions for our study. Reponses to these comments are provided below.**

Methodology:

The authors used SSP emission scenarios for foreign countries and another different set of emission scenarios, developed by Tong et al. (2020) and Cheng et al. (2021b), for China. I understand the motivation of combining these two datasets. But would it be helpful to show the results in the supplement on using SSP emission scenarios for both foreign countries and China? Will those results be broadly consistent with the main findings reported here?

**Response: The referee suggested PM$_{2.5}$ simulations with SSP emission scenarios for both foreign countries and China have actually been conducted in Cheng et al. (2021a). They found that Tong et al. (2020) and Cheng et al. (2021b) emissions were developed specifically to meet China's recent climate goals (i.e., 2030 carbon peak and 2060 carbon neutrality) and used parameters (i.e., emission factors, energy use, etc.) to reflect the most up-to-date pollutant control policies (clean air action since 2013) and technologies in China. On the contrary, SSP emissions use air pollution scenarios from global databases (e.g. the GAINS database and ScenarioMIP for CMIP6) that do not reflect the clean air policies and air quality improvements in China since 2010 (Cheng et al., 2021b). As a result, simulations with Tong et**

al. (2020) and Cheng et al. (2021b) emissions could better capture China's $PM_{2.5}$ concentration decline during 2015–2019 than those driven by SSP emissions (Cheng et al., 2021a). With the effects of current-year bias and inadequate considerations of pollution control policies, $PM_{2.5}$ projections in 2030 and 2050 with SSP mitigation scenarios are 42%–48% (9–13 µg m$^{-3}$) and 59%–73% (8–12 µg m$^{-3}$) higher than projections with Tong et al. (2020) and Cheng et al. (2021b) (Cheng et al., 2021a). Thus, there are considerable differences in emissions and corresponding $PM_{2.5}$ concentrations under the two kinds of scenarios, which warrants our choice of region-specific emission scenarios.

We have revised the main text on page 5 to clarify the reasons of our choice: "Future scenarios of anthropogenic pollutant emissions for China were developed by Tong et al. (2020) and Cheng et al. (2021b). We adopt these emissions instead of SSP-RCP emissions because they were developed specifically to meet China's recent climate goals (i.e., 2030 carbon peak and 2060 carbon neutrality) and used parameters (i.e., emission factors, energy use, etc.) to reflect the most up-to-date pollutant control policies (i.e., clean air action since 2013) and technologies in China. Their emissions have been suggested to better capture China's $PM_{2.5}$ concentration decline during 2015–2019 than those driven by SSP emissions (Cheng et al., 2021a)."

Analysis:

Fig. 3 suggests that East China is a hotspot of transboundary pollution-induced PM2.5 in all scenarios investigated. But it is not discussed in the paper. I suggest conducting deeper analysis and/or adding more discussion on this area. You can potentially compare these results with those for Yunnan and Xinjiang.

Response: Thanks for the suggestion. We included more discussion on the mechanism of transboundary pollution over North China on page 9 as the following:

"Xu et al. (2023) found that foreign-transported NMVOCs could enhance the atmospheric oxidizing capacity and facilitated the oxidation of Chinese nitrogen oxides ($NO_x$) to form nitrate over North China. This feature persists in the future. Due to chemical interactions, North China is the region that is the most strongly affected by transboundary pollution in most scenarios. In 2030, transboundary pollution contributes 3 to 10 µg m$^{-3}$ $PM_{2.5}$ concentrations to North China, even under China's carbon-neutral scenario (Fig. 3b and Fig. 3i-l), greatly increasing the difficulty for this region to achieve the 35 µg m$^{-3}$ goal. Further reduction of transboundary pollution on North China requires the reduction of not only foreign anthropogenic emissions but also China's domestic emission."

We also compared the influence of foreign anthropogenic emissions on North China and Yunnan province by sector and source in Fig. S4 and Fig. S5. Discussions are included on page 10 line 33-40.

Related to that, I wonder where these hotspots of transboundary PM2.5 originate from. This requires further analysis using, for example, back-trajectory models or sensitivity tests with regional emissions, and may therefore be out of scope of this study. But it could be helpful to add some discussions on that if relevant papers are available.

**Response: Thanks for the suggestion. We conducted a sensitivity simulation to understand whether transboundary pollution came from countries within or outside our Flex-Grid domain (the entire Asia, eastern Europe and eastern Africa), as shown in Fig. S5. We added the discussion on the sources of transboundary pollution on page 10 as the following:**

**"Transboundary pollution to China could arise from emissions of countries within and outside the Flex-Grid domain of the model (11° S–60° N, 30°–150° E, including the entire Asia, eastern Africa and eastern Europe). Thus, we conduct a sensitivity simulation driven by China's current-policy and foreign countries' SSP370 emissions in 2030 and 2060 as an example to explore the relative importance of anthropogenic emissions from countries within and outside the Flex-Grid domain for transboundary pollution to China. Fig. S5 shows that anthropogenic emissions from countries within the Flex-Grid domain dominate the transboundary pollution to China (94%), and their contributions remain in 2030 and 2060. Contributions of anthropogenic emissions from countries outside the Flex-Grid domain is larger in April and October when westerly winds prevail and wet deposition is not as strong as in July (Jiang et al., 2013; Leibensperger et al., 2011; Ni et al., 2018). North China is more influenced by emissions outside the Flex-Grid domain (13% in April and 24% in October) than China on average is (7% in April and 13% in October), indicating that North China is more influenced by emissions outside Asia, such as Europe and North America. Leibensperger et al. (2011) have also found that $NO_x$ emission in the US could lead to an enhancement of $PM_{2.5}$ over North China. Yunnan province is almost completely (≥ 99%) affected by countries within the Flex-Grid domain."**

Figures:

North China is not explicitly defined in the text or figures. Line 25 on Page 9 states that North China is outlined in Fig. 3c-o but I cannot see that information. Please clarify.

**Response: Thanks for pointing this out. We added "North China is outlined by thick black lines" to the caption of Fig. 3.**

**Authors' Response to Comments from Editor**

Reviewer #1 brought to my attention that there is a high degree of copying text in the Methods section from the authors' previous 2023 ACP paper. The similarity report that ACP produces did not notice the strong similarity in this text, and if it had, I would have asked for it to be changed or removed before the Discussion paper was posted. I agree with Reviewer #1 that it shouldn't be necessary, and that it is inappropriate, to copy all of this text on how the model and emissions were set up. I suggest shortening this discussion to recap the main points, while referring to the previous paper for more details.

**Response: We sincerely apologize for copying the text in the Method section from our previous paper. We have revised the text accordingly as stated in our response to Referee #1.**

References:

[revised manuscript text omitted]